# Chondrocyte Isolation from Loose Bodies—An Option for Reducing Donor Site Morbidity for Autologous Chondrocyte Implantation

**DOI:** 10.3390/ijms24021484

**Published:** 2023-01-12

**Authors:** Martin Textor, Arnd Hoburg, Rex Lehnigk, Carsten Perka, Georg N. Duda, Simon Reinke, Antje Blankenstein, Sarah Hochmann, Andreas Stockinger, Herbert Resch, Martin Wolf, Dirk Strunk, Sven Geissler

**Affiliations:** 1Center for Regenerative Therapies (BCRT), Berlin Institute of Health at Charité—Universitätsmedizin Berlin, Augustenburger Platz 1, 13353 Berlin, Germany; 2Julius Wolff Institute, Berlin Institute of Health at Charité—Universitätsmedizin Berlin, Augustenburger Platz 1, 13353 Berlin, Germany; 3Centrum für Muskuloskelettale Chirugie (CBMSC), Charité—Universitätsmedizin Berlin, Corporate Member of Freie Universität Berlin and Humboldt-Universität zu Berlin, Augustenburger Platz 1, 13353 Berlin, Germany; 4Med Center 360 Degree Berlin, Kieler Straße 1, 12163 Berlin, Germany; 5Wyss Institute for Biologically Inspired Engineering at Harvard University, Boston, MA 02138, USA; 6Cell Therapy Institute, Spinal Cord Injury and Tissue Regeneration Center Salzburg (SCI-TReCS), Paracelsus Medical University (PMU), 5020 Salzburg, Austria; 7Department of Traumatology, Clinic Schaerding, 4780 Schaerding, Austria; 8Department of Traumatology, Paracelsus Medical University (PMU), 5020 Salzburg, Austria; 9Berlin Center for Advanced Therapies (BECAT), Charité Universitätsmedizin Berlin, 13353 Berlin, Germany

**Keywords:** cartilage, human chondrocytes, loose body, regeneration, ACI, OCD, BMSC

## Abstract

Loose bodies (LBs) from patients with osteochondritis dissecans (OCD) are usually removed and discarded during surgical treatment of the defect. In this study, we address the question of whether these LBs contain sufficient viable and functional chondrocytes that could serve as a source for autologous chondrocyte implantation (ACI) and how the required prolonged in vitro expansion affects their phenotype. Chondrocytes were isolated from LBs of 18 patients and compared with control chondrocyte from non-weight-bearing joint regions (*n* = 7) and bone marrow mesenchymal stromal cells (BMSCs, *n* = 6) obtained during primary arthroplasty. No significant differences in the initial cell yield per isolation and the expression of the chondrocyte progenitor cell markers CD44 + /CD146+ were found between chondrocyte populations from LBs (LB-CH) and control patients (Ctrl-CH). During long-term expansion, LB-CH exhibited comparable viability and proliferation rates to control cells and no ultimate cell cycle arrest was observed within 12 passages respectively 15.3 ± 1.1 mean cumulative populations doublings (CPD). The chondrogenic differentiation potential was comparable between LB-CH and Ctrl-CH, but both groups showed a significantly higher ability to form a hyaline cartilage matrix in vitro than BMSC. Our data suggest that LBs are a promising cell source for obtaining qualitatively and quantitatively suitable chondrocytes for therapeutic applications, thereby circumventing donor site morbidity as a consequence of the biopsies required for the current ACI procedure.

## 1. Introduction

The most common disorder in which loose bodies occur is osteochondritis dissecans (OCD). It might result most commonly in pain, predominantly in late stages, due to physical activities. Other symptoms include swelling, joint locking or a decrease in the range of motion. Overall, there are no general typical clinical signs [1]. This disease predominantly affects adolescent patients between 12 and 19 years of age. With a ratio of nearly 4:1, it mainly affects male patients and primarily the medial femoral condyle of the knee [2,3,4,5]. Even 130 years after its first description by Franz Koenig [6], the underlying pathomechanism of the disease is still not fully understood, but micro-fractures, microtrauma and/or ischemia are thought to be the leading causes [7]. In recent years, a continuous rise in the incidence rate of OCD has been observed in young female patients, which has been associated with the increase in sports activities [4,8]. Higher prevalence in adolescent still in growth stage supports the theory of increased disposition to microfractures/traumata and the need for a minimally invasive therapeutic approach.

OCD at stage IV of the International Cartilage Regeneration & Joint Preservation Society (ICRS) knee score have a completely detached fragment (loose body, LB), causing persistent blockage and pain [9,10]. This necessitates surgical intervention to restore the bone–cartilage compartment. Among current treatment options, autologous chondrocyte implantation (ACI) is considered the most effective method to achieve healing of the damaged tissue area. During this procedure, a cartilage biopsy is first taken from non-weight-bearing areas of the knee joint, which is used to isolate, and ex vivo propagate chondrocytes, which are then re-implanted into the weight-bearing defect area.

A major drawback of this autologous approach is a possible donor site morbidity, which can lead to persistent impairment [11,12,13]. This is of particular concern because most patients are young and highly active. Transplantation of allogenic chondrocyte is an alternative, but is not yet in clinical practice due to the associated ethical concerns, risk of disease transmission and possible immune-mediated graft rejection [14].

Some studies have shown that LB fragments contain viable chondrocytes that could be used for ACI [15,16,17]. However, the effects of the necessary extensive in vitro expansion on the function and chondrogenic phenotype of the isolated cells have been neglected or insufficiently quantified [18,19]. The aim of this study was to systematically evaluate LB-derived chondrocytes as a potential cell source for ACI treatment, especially with regard to the influence of long-term in vitro expansion on their re-differentiation potential compared to chondrocytes from unloaded healthy knee areas. Bone marrow-derived stromal cells (BMSCs), which also have significant chondrogenic differentiation potential and are discussed as an alternative cell source for cartilage therapies, were used as an additional control [20]. BMSCs are considered as precursor cells of chondrocytes and are currently the gold standard for regenerative therapies in the musculoskeletal field, e.g., as a starting material or in a combination with chondrocytes [21,22,23]. Stromal cells from other tissue have also attracted much attention; however, their regenerative properties are mainly based on paracrine effects rather than differentiation into functional tissue cells, especially in the context of cartilage regeneration [24].

## 2. Results

### 2.1. Histological Analysis of a Loose Body

A qualitative histological analysis of the native loose body specimens shows a typical tri-zone of organization of cartilage tissue (Figure 1 and Figure A1). This starts with the outer superficial zone, e.g., clearly visible in the elastic staining and the outstretched chondrocytes parallel to the border. It is followed by the transitional zone with rounded chondrocytes and a higher collagen (Movat’s Pentachrome) content. The highest proteoglycan (Alcian blue) and cartilage (Safranin O/Movat’s Pentachrome) prevalence are seen in the inner radial zone [25]. Nuclei are seen well distributed in all zones in bright red (Alcian blue), blue black to purple (Hematoxylin & Eosin) or brown to black (Elastica) depending on the staining. Strikingly, in the Movat’s Pentachrome staining some outer regions of the LB still show osteoid in red. Cytoplasmic parts stained in pink to red (Hematoxylin & Eosin), brick red (Masson–Goldner Trichrom) and yellow (Elastica). Connective tissue is seen in red (Elastica), yellow to red (Movat’s Pentachrome) and green (Masson–Goldner Trichrom). Tissue in general is stained in green via Safranin O/Fast Green. All histological evidence exhibits a healthy and highly organized cartilage structure.

### 2.2. Isolation and Long-Term Proliferation

We deliberately chose not to completely digest the cartilage samples (e.g., overnight), but instead performed a partial digestion of the fragments to facilitate outgrowth and viability. Outgrowth at passage 0 (P0) took 13 ± 6 days for the LB_CH group and 21 ± 4 days for the CTRL-CH samples (*p* = 0.09). All isolated cells showed a small, spindle-shaped, and fibroblast-like morphology corresponding to that of dedifferentiated chondrocytes (Figure A2) [26]. Cell yield after P0 was comparable between LB-CH and CTRL-CH groups, with a mean cell number/isolation of 2.5 ± 1.0 *×* 10^5^ and 2.7 ± 1.2 *×* 10^5^, respectively. Isolated chondrocyte populations from LB and CTRL donors showed comparable viability (85.8% ± 4.6 and 87.4% ± 6.2, respectively) in P1 and achieved similar total cell number (5.3 ± 2.1 *×* 10^6^ and 5.6 ± 2.5 *×* 10^6^ cells/isolation, respectively). Regardless of the origin, the isolated cells maintained the fibroblast-like morphology (Figure A2). It is worth mentioning that the yield could be further increased, either by multiple primary seeding or by reseeding the fragments after the initial digestion. In this study, fragments were separated from the single cell suspension using a cell strainer and discarded after the first passage.

For in vitro expansion, chondrocytes were seeded at 33,333 cells/cm^2^ (2.5 × 10^5^ per T75 flask) in each passage and subcultured for a constant duration of seven days before being trypsinized and transferred to the next passage. LB-CH and CTRL-CH exhibited similar proliferation rate and viability throughout the whole expansion process (Figure 1a). All chondrocyte cultures were maintained for up to 12 passages without ultimately reaching the state of cell cycle arrest. Additionally, during the whole long term culture process cell have always been in a subconfluent state. No significant differences in the number of cumulative PD were observed between LB-CH (15.3 ± 1.1) and CTRL-CH (17.3 ± 3.6) (PD and CPD, Figure 2a).

To determine the amount of chondrocyte-derived progenitor cells (CDPCs) in each cell isolation, the expression of cell surface marker CD44 and CD146 were quantified by flow cytometry [27,28,29]. For this purpose, cells were seeded at fourfold higher densities (2000 cells/cm^2^) than low-density seeding conditions to avoid active selection and proliferation of CDPCs [30]. The focus was on the initial purity, the percentage of CDPCs and their change due to long-term cultivation effects under standard conditions (Figure 2b). LB-CH and CTRL-CH show a high expression of the CD44 and CD146 marker in early in vitro cultures (P2: 44.6% ± 13.5 and 54.5% ± 20.2, respectively) and no significant difference were observed between both groups (*p* = 0.165). The number of CD44 and CD146 double positive cells decreased with increasing cultivation time, so that the amount of CDPCs in both group at P12 were only 30.9% ± 16.6 (LB-CH) and 33.7% ± 10.1 (CTRL-CH).

### 2.3. Apoptosis Rate, Short-Term Proliferation, and Metabolic Activity at Early and Late Passage

To assess the number of apoptotic cells in early (P2) and late (P12) LB-CH, CTRL-CH, and BMSC cultures, caspase 3/7 activity assays were performed. This analysis revealed significantly higher caspase 3/7 activity levels in early BMSC cultures compared to the corresponding early LB-CH (*p* = 0.0004) and CTRL-CH (*p* < 0.0001) populations (Figure 3a). This difference diminished during in vitro culture, so that the different groups showed a comparable apoptosis rate in P12.

To further characterize the impact of in vitro expansion on LB-CH and CTRL-CH in comparison to BMSCs, the proliferation capacities of early and late cultures were determined (Figure 3b,c). Early cultures in P2 achieved comparable population doubling rates within nine days (LB-CH: 1.7 ± 0.3, CRTL-CH: 1.5 ± 0.1, and BMSC: 1.6 ± 0.4). The population time [h] showed also no significant difference between the early chondrocyte and BMSCs cultures (LB-CH: 56.6 h ± 1.4; CRTL-CH: 57.3 h ± 0.5, and BMSCs: 56.5 h ± 2.1). In line with the low apoptosis rate, late chondrocyte and BMSC cultures in P12 showed comparable PD and PD time (LB-CH: 56.9 h ± 1.7 with 1.5 ± 0.3 PD, CTRL-CH: 56.2 h ± 1.7 with 1.7 ± 0.4 PDs, and BMSCs: 58 h ± 1.5 with 1.3 ± 0.3). Subsequently the metabolic activity of the cultures was determined using a PrestoBlue™ Cell Viability assay (Figure 3d). This analysis revealed that the metabolic activity is significantly lower in late passage LB-CH (*p* = 0.0118) and BMSC cultures compared to their early passage counterparts, suggesting increased cellular senescence. However, no significant differences were observed between the three different groups at the same passage.

### 2.4. Chondrogenic Re-Differentiation

The chondrogenic phenotype and (re-)differentiation potential of chondrocytes and BMSCs was tested using a pellet culture assay and differentiation was induced with specific chondrogenic induction medium (ChM). Cell pellet cultures in ChM without addition of TGF-ß served as negative control.

Alcian blue staining for qualitative detection of sulfated glycosaminoglycans (such as hyaluronic acid or chondroitin sulfate) showed pronounced chondrogenic differentiation of all TGF-ß-treated cultures compared to the corresponding negative control (without TGF-ß). Both LB-CH and CRTL-CH exhibited a similar Alcian blue staining and formed larger pellet cultures then BMSC cultures (Figure 4a,b).

Immunohistochemical staining of collagen type II confirmed successful chondrogenic (re-)differentiation of TGF-ß-stimulated LB-CH pellets cultures with a three-zone separation compared to the negative controls. The outermost layer reflecting the superficial zone and is clearly distinguishable from the underlying transition/radial zone. The inner part of the pellets appeared to be less dense and structured, as did the other two zones (Figure 4c).

To quantitatively assess the chondrogenic (re-)differentiation, the total proteoglycan content was determined (Figure 4d,e). This analysis revealed that LB-CH cultures in P2 produced significantly more chondrogenic matrix (340.1 ± 194.1 µg PG per mg total protein) as corresponding BMSC cultures (113.5 ± 53.6 µg PG per mg total protein, *p* = 0.004), but no difference to CTRL (169 ± 122.7 µg PG per mg total protein, *p* = 0.1058) (Figure 4d). Late LB-CH and CTRL-CH cultures in P12 were also successfully (re-)differentiated and produced a similar amount of chondrogenic matrix (LB-CH: 263.3 ± 142.3 µg and CTRL-CH 300.5 ± 101.8 µg PG/mg protein) (Figure 4e).

Quantitative PCR was used to assess mRNA expression of essential chondrogenic ECM components in early passage cultures after (re-)differentiation. Gene expression was normalized to beta-2-microglobulin as a housekeeper gene, and actin (ACTB) and hypoxanthine-guanine phosphoribosyltransferase (HPRT) served as additional controls (Figure 5). BMSC cultures without TGF-ß served as reference to define baseline expression values of the undifferentiated cell state. In line the total proteoglycan content, LB-CH exhibited significantly higher aggrecan and collagen II mRNA expression compared to BMSC (ACAN *p* = 0.0263; COL II *p* = 0.0104). While no significant difference in mRNA expression were found between LB-CH and CTRL-CH, both groups showed significant higher SOX9 expression compared to BMSCs. In contrast, the mRNA expression of the osteochondral marker collagen X was significantly higher in BMSCs compared to both LB-CH and CTRL-CH (*p* = 0.0108 and *p* = 0.048). Similarly, the expression of MMP13 was elevated in BMSCs compared to both chondrocyte cultures, but did not reach statistical significance.

In summary, these analyses showed that LB-CH exhibited a similar chondrogenic phenotype as the corresponding CTRL-CH, and maintained their differentiation capacity even after prolonged in vitro culture.

## 3. Discussion

Although ACI for e.g., OCD treatment is hitherto the optimal solution, it is rather ironic to apply additional damage to a patient’s cartilage to harvest cells to heal the primary cartilage defect. Especially given the fact that a massive cell source (the loose body) is discarded in that process. Even after longer time periods of detachment the in vivo survival and supply of nutrition for LBs is archived via the synovial fluid [31] and actively by the subchondral bone as shown in animal studies [32,33]. Some other publications report that it is not possible to establish chondrocyte cultures from all LBs, especially in patients whose disease onset dates back more than one year [34]. We were able to isolate chondrocytes from all LBs obtained and only in two exceptional cases had to discard isolated cells because of early bacterial contamination. Compared with chondrocytes from healthy cartilage regions, the isolated cells showed an excellent proliferation rate, cartilage marker expression, and chondrogenic matrix production and were even superior to human BMSCs. The reason for the difference between our results and those of others may be that a longer overnight digestion was chosen in these studies, which in our experience may lead to decreased viability [34]. In addition, the isolated cell solutions are often subsequently filtered, which may also lead to damage or destruction of the cells by shear forces, especially after the prolonged digestion process.

The limitation of our study is, of course, the origin and size of the control (CTRL) group. Even though we isolated the control cells from non-weight-bearing healthy cartilage areas, a perfect control cohort would have included biopsies derived from each LB patient individually, which is not feasible considering ethical criteria and would entail additional damage to the patients (Table 1). Although appropriate biopsies for standard ACI were also taken from the patients in the LB group and cells were isolated under GMP conditions, we could not include them in our research as they were completely required for the treatment according to the routine protocol. The aim of our work was to circumvent the collection of cartilage biopsies from intact (presumably healthy) non-loaded areas of the knee joint by using cells from the LB. Other treatment options are of course available for these patients (e.g., microfracture, etc.), but several clinical trials (including phase III) show that ACI is superior to these alternative procedures [35]. 

A first insight into the feasibility of using LB as a cell source for ACI was delivered by our qualitative histological analysis of native loose bodies and their corresponding synovial pocket (Figure 1 and Figure A1). These revealed an intact and healthy cartilage structure of the loose bodies. After successful isolation, in high quantities, the long-term cultivation and analysis were conducted to simulate long-term survival and the impact of in vitro aging as an approximation for an in vivo survival. This represents the foundation for any successful long-term therapy. Surprisingly, we did not detect growth arrest in long-term settings, or any differences between the LB and CTRL group (Figure 2a).

Analysis of CD44, hyaluronan receptor, marker expression reflects the purity and potential of the isolated cells [36,37]. The high rate of CD44/CD146 positive cells shows that the isolation protocol yields high potential chondrocytes. As described by Jiang and colleagues [30] we checked early and late passages of LB and CTRL chondrocytes for chondrocyte-derived progenitor cells via the expression of the CD146 marker (Figure 2b.). Positive cells are postulated to have a greater regenerative potential as compared to CD146 negative cells. Surprisingly we saw that in the CTRL cohort there is a significant drop in positive cells (20.8% *p* = 0.037) between passage 2 and 10 but not in LB (13.7% *p* = 0.058). In respect of this, the detrimental effect of in vitro aging correlates with our data of re-differentiation and its markers. For BMSC the high impact of in vitro aging was shown by our group before [38], surprisingly in low passage (≤4) chondrocytes show only an in vitro difference based upon their donor age [39].

We then went on to confirm cell survival with a specific focus on long time-culture periods (P12) and in short-term culture (P2). This was carried out via checking for the apoptosis hallmark cascade of caspase 3/7 (Figure 3a). Here BMSC demonstrate a significant higher base level in P2 compared to LB and CTRL and to a lesser, non-significant, extend in the high passage (P12). No group changed significantly from P2 to P12, which supports the long-term data as well as all microscopical observations.

A crucial timeframe for therapies are always the first passages (P1–P2) for massive ex vivo expansion followed by the re-implantation procedure. We therefore checked extensively for short-term effects such as population doublings and time and metabolic activity, as a marker for viability. These were conducted over nine days in the early passages as well as in the late passages for long-term representation. No significant differences were observed in early or late passages. Surprisingly, BMSC had the longest population doubling time of all three groups in the late passage (Figure 3b,c). Metabolic activity as an indicator for viability was highest in early passage BMSC and the only significances were found between early and late passage, as expected, likely due to in vitro aging effects (Figure 3d).

Proteoglycan measurement after the re-differentiation process was carried out again in early and late passages. The only significance in proteoglycan levels could be detected between the LB and BMSC group in the early passage (P2) (Figure 4d,e). The high patient’s variance in the LB cohort is most likely due to the age and loose body variety (size, duration disease state, pathophysiology etc.) in this group (Table 1).

As quality has to be chosen over quantity, in our study ex vivo expansion was never a limiting factor, and the re-differentiation process yielded very good results for LB in the crucial early passage. Additionally, we histologically confirmed the quality after the re-differentiation via Alcian blue (proteoglycans) and collagen II staining (Figure 4a–c). Overall, both cell sources were viable and capable of proliferation even until P12 (Figure 2 and Figure 3).

The significantly higher aggrecan and SOX9 mRNA levels of LB correlate with the higher proteoglycan levels of early chondrogenic markers, which indicate an advanced progression in the differentiation process for LB and CTRL cells in comparison to BMSC (Figure 5) [40]. MMP13 levels in BMSC indicate an inferior quality of cartilage in terms of hypertrophy. Evidence suggests that BMSCs tend to form hypertrophic cartilage [41]. Our study supports this idea as we see a high COL X/COL II ratio and higher MMP13 levels in BMSC, which reflects transient cartilage. This seems logical if one considers the “true” nature of BMSCs as a pre-state for bone (regeneration). So the value of a BMSC-based therapy in cartilage regeneration might be by default inadequate in terms of endochondral ossification [22]. An allogenic chondrocyte therapy seems much more promising in producing the proper kind of cartilage. Furthermore, the in vitro performances of BMSCs were inferior within most parameters of our study. Our results support the conclusion that BMSCs are less suitable for cartilage reconstruction therapies and that cells of chondrogenic origin should be used for cartilage defect therapy.

Our data support the usage of loose bodies as a cell source under specific isolation conditions (partial digestion and sub-standard seeding densities), which leads to an excellent ex vivo amplification. This underlines the importance of choosing the right cell source for the right therapeutical approach to achieve the most beneficial outcome for the patient with the lowest drawbacks.

## 4. Materials and Methods

### 4.1. Ethics Statement

Patient recruitment and sample harvest was approved by the local ethical committee (EA2/089/20), and all donors gave written informed consent.

### 4.2. Chondrocyte Isolation and Culture

Loose bodies of 18 patients (mean age 31.2 ± 11.2 years), which were not eligible for re-fixation of the fragment, were included in this study. Arthroscopy of the knee was performed, and the detached osteochondral fragment was removed. Chondrocytes from healthy non-weight-bearing cartilage regions of patients undergoing a knee TEP (control, CTRL) served as references (*n* = 5; mean age 64.2 ± 7.8 years). All specimens were harvested under sterile conditions and stored in sterile tubes filled with phosphate-buffered saline (PBS) until further use. Processing started immediately after cartilage harvest. Therefore, isolated cartilage was cut into small fragments of approximately 2–4 mm diameter and digested sequentially, first using 350 U/mL pronase E (Sigma Aldrich, St. Louis, MO, USA) and then 100 U/mL collagenase type II (Biochrom, Berlin, Germany), each 1 h at 37 °C, under continues agitation, respectively. Cartilage fragments were centrifuged (300× *g*, 15 min), re-suspended in cell culture medium and seeded on culture flasks. The cell culture medium was Dulbecco’s modified Eagle’s medium (DMEM) low glucose (Sigma Aldrich, D5546, USA) with 10% fetal calf serum ‘superior’ (Biochrom, S0615, Germany), 2 mM GlutaMAX™ (Gibco, Thermo Fisher Scientific, Waltham, MA, USA), penicillin (10 U/mL)/streptomycin (10 µg/mL) (Biochrom, Germany). Sub-cultured cells were seeded at 2400 cells/cm^2^ in cell culture flasks. Cell number and viability of chondrocytes and BMSCs were determined using the cell counter CASY TT (OLS OMNI Life Science, Bremen, Germany). All assays were carried out with primary cells either in passages 2–4 or 10–12, respectively.

### 4.3. BMSC Isolation and Culture

Primary age- and gender-matched, to the CTRL group, human bone marrow-derived BMSCs (*n* = 6; mean age 57.7 ± 22.1 years), isolated as described [42] were used as an additional reference. The supplemented DMEM cell culture medium was the same as for chondrocytes above. Cells were counted by using CasyTT for standard cell culture and cryopreserved until further use.

### 4.4. Chondrogenic Differentiation

Chondrocyte phenotype was determined under re-differentiation. For chondrogenic re-differentiation, a pellet culture assay with chondrogenic differentiation media (ChM) was performed for 21 days in a 96 deep-well format as described elsewhere [43,44]. Briefly, 3 × 10^5^ cells per pellet were used, and cultured in ChM composed of high glucose DMEM including L-glutamine (Sigma Aldrich, D6546, USA), dexamethasone (0.1 µM, Sigma Aldrich, D2915, USA), L-ascorbic acid (50 µg/mL, Sigma Aldrich, A8960, USA), L-proline (40 µg/mL, Sigma Aldrich, P5607, St. Louis, MO, USA) sodium pyruvate (0.1 mg/mL, AppliChem, A4859, Darmstadt, Germany) ITS (6.25 µg/mL, Sigma Aldrich, I1884, St. Louis, MO, USA), BSA (1.25 mg/mL, Sigma Aldrich, A9418, St. Louis, MO, USA), linoleic acid (5.35 µg/mL, Sigma Aldrich, L1012, St. Louis, MO, USA), penicillin (10 U/mL)/streptomycin (10 µg/mL) (Biochrom, A2213, Berlin, Germany) and recombinant human TGF-ß1 (10 ng/mL, Peprotech, 100-21-5, Cranbury, NJ, USA). Negative controls were cultured in the same medium but without TGF-ß1. Media change was performed every 3–4 days. Chondrogenic phenotype was determined by proteoglycan assay, type II collagen immunostaining, Alcian blue staining, as well as by expression of type II collagen, type X collagen, MMP13, Sox9 and aggrecan mRNAs.

### 4.5. Viability and Proliferation

Cell viability was measured by using PrestoBlue^®^ cell viability reagent (Life Technologies, Carlsbad, CA, USA). Proliferation rates and cell population doublings (PD) were acquired by using a CyQUANT^®^ cell proliferation assay kit (Life Technologies, USA). Assays were conducted according to the manufacturer’s instructions. In brief, 2500 cells/cm^2^ were seeded per 48-well for each individual time point, starting 24 h after seeding marked the time points zero (d0), day three (d3), day six (d6) and day nine (d9). Media change was performed every three days. All determinations were executed in triplicates or quadruplicates using a multimode microplate reader (m200 pro, Tecan, Maennedorf, Switzerland). Relative fluorescence units (RFU) were used to calculate population doublings and time. Long-term cumulative population doublings were determined by plating defined cell numbers per cm^2^ (2.5 × 10^5^ per T75 cell culture flask). After seven days, cells were trypsinized and counted with a cell counter CASY TT (OLS OMNI Life Science, Bremen, Germany).

### 4.6. RNA Extraction and qPCR

After 21 days of chondrogenic differentiation, two pellets were pooled with 250 µL TRIzol^®^ (Thermo Fisher Scientific, USA) within a 2.0 mL screw cap micro tube (Sarstedt, Nuembrecht, Germany). Three 2.8 mm and twenty-five 1.4 mm Precellys ceramic beads (Peqlab, Erlangen, Germany) were included. A Minilys homogenizer (Peqlab, Erlangen, Germany) was used for mechanical decomposition of the pellets (3 × 1 min, max speed). Afterwards, the solution was transferred to a new 1.5 mL Eppendorf tube and each screw cap micro tube was washed three times with 250 µL TRIzol^®^. RNA was than extracted from each pool (1 mL) according to the TRIzol^®^ manufacturer’s instructions. Glycogen (10 µg) was used to increase the yield and visibility of the RNA pellet, which was finally reconstituted in 20 µL RNase-free water. RNA concentration was determined using a NanoDrop 1000 (Thermo Fisher Scientific, USA). RNA (500 ng) was then transcribed with iScript™ cDNA synthesis kit (Bio-Rad, Hercules, CA, USA) according to the manual. Next, qPCR was performed using the LightCycler^®^ 480 SYBR Green I Master Mix (Roche, Basel, Switzerland) with 5 ng cDNA and 200 nM primers (Table A1). All qPCR runs were executed on a LightCycler^®^ 480 II machine (Roche; Switzerland). Further analysis was carried out with a LightCycler^®^ 480 SW 1.5.1 software and the fit point analysis method with a fixed threshold of 1.0 for all samples. Normalization was conducted by relating the genes to the housekeeping gene beta-2 microglobulin (b2 MG) yielding ∆Ct. Afterwards each gene was normalized to the corresponding gene of the TGF-ß negative BMSC culture resulting in the ∆∆Ct.

### 4.7. Proteoglycan Determination

As described before by Davis et al. [43], we extracted proteoglycan (PG) from two pooled pellets of each condition, as in the RNA preparation, except that 150 µL proteoglycan extraction buffer (PEB) was used instead of TRIzol^®^. Immediately after adding the DMMB assay reagent, absorption at 516 nm was acquired on a Tecan microplate reader. PG data was normalized to the total protein content of each sample, determined with a Coomassie (Bradford) protein assay kit (Thermo Fisher Scientific, USA) according to the manual.

### 4.8. Apoptosis

Apoptotic cells were identified by capase−3/7 activity (Caspase-Glo 3/7 assay; Promega, Madison, WI, USA) as described [45] and according to manufacturer’s instructions and normalized to the fluorescent measurement from the CyQUANT^®^ cell proliferation assay kit (Life Technologies, Carlsbad, CA, USA).

### 4.9. Histology and Immunohistology

Four µm sections of formalin-fixed, paraffin-embedded tissue were used for analysis. Automated hematoxylin and eosin (H&E) staining was carried out in a linear slide stainer (Leica ST4040) using Mayer’s Haemalaun (Merck; 1.09249.2500, Rahway, NJ, USA) and Eosin Y (Merck; 1.15935.0100) [46]. Movat’s Pentachrome (Verhoeff) staining was performed according to the manufacturer’s protocol (Morphisto 12061, Offenbach, Germany). Safranin O/Fast Green staining was performed according to a standard protocol. Briefly, after deparaffinising and hydrating the paraffin sections, they were incubated in Weigert’s Iron Hematoxylin (Hematoxylin, 517-28-2; Ferric Chloride 7705-08-0, Merck) for five minutes and afterwards washed in distilled water three times. The sections were then differentiated in 1% acid–alcohol for 10 s and rinsed in distilled water three times. Subsequently the sections were incubated for one minute in 0.2% Fast Green (Morphisto 10267), 15 s in 1.0% acetic acid, followed by 30 min in 1.0% Safranin O (477-23-6, Merck). Afterwards the slides were briefly rinsed in 96% ethanol and dehydrated with two changes of 96% ethanol and 100% ethanol. Afterwards the sections were washed in acetic acid *n*-butyl ester (P036.1 Roth) and mounted. Cartilage (proteoglycan specific) stains bright red and the backgrounds stains in green [47]. For Masson–Goldner trichrome and elastica staining, kits were used according to the manufacturer’s recommendations (Morphisto 12043, Morphisto 14604) [48]. Alcian blue/nuclear fast red staining was performed according to a standard protocol, see below. Slides were scanned automatically in 20× or 40× magnification using the VS-120-L Olympus slide scanner 100-W system. Pictures were processed using the Olympus VS-ASW-L100 program and OlyVIA software.

One pellet of each differentiation was paraffin-embedded. Slices (3 µm) of the mid-section were further processed. One set was stained with Alcian blue for glycosaminoglycans, as described by Yang et al. [49]. Counterstaining was performed with nuclear fast red. The other set of slices was processed for collagen type II immunohistochemistry. Briefly, samples on slides were deparaffinized, treated with hyaluronidase (0.02% in PBS) for 2 h at 37 °C, digested with pepsin (0.1% in 0.01 M HCl) for 30 min at 37 °C, washed, and then blocked with 2% BSA/2% normal goat serum in PBS. Primary collagen II antibody (Quartett, Potsdam, Germany, Clone 2B1.5) was diluted 1: 50 in 2% BSA/2% normal goat serum in PBS and incubated overnight at 4 °C. After washing, goat anti-mouse Alexa Fluor^®^ 488 (Thermo Fisher Scientific, USA) secondary antibody was added (1: 200, in PBS with 2% BSA/2% normal goat serum) for 1 h at RT. Nuclei were stained afterwards with 4′,6-diamidin-2-phenylindol (DAPI 1: 1500 in distilled water) for 10 min. Microscope slides were covered with Fluoromount G (Southern Biotech, Birmingham, AL, USA) and glass slides. All pictures were taken with a Zeiss Axio Scope (Oberkochen, Germany) microscope.

### 4.10. FACS Analysis for CD146 and CD44

Cells were trypsinized at ~90% confluency for the given low and high passage. After washing, cells were stained with 20 µL/1 × 10^6^ cells CD146-PE (BD #550315) and CD44-FITC (BD #347943), respectively. Cells were fixated and acquired on a BD LSRFortessa™. A minimum of 5 × 10^5^ total events were acquired for every sample. Analysis was performed with FlowJo^®^ (FlowJo, LLC, Ashland, OR, USA).

### 4.11. Statistical Analysis

All statistical analysis was performed with the software Prism 9.41 (Dotmatics). One-way ANOVA with multiple comparisons and Tukey compensation was utilized for all analysis, except for the CPD analysis, which was performed using a two-way ANOVA with multiple comparison and Sidak compensation for more robustness.

## 5. Conclusions

Loose bodies show no impaired behavior compared to primary cartilage. This might make them a promising tissue source and efficient alternative for harvesting otherwise intact cartilage for ACI based upon our results, which would also circumvent the need of further biopsies and donor side morbidity. The latter is of high importance considering the immense prevalence of e.g., osteochondritis dissecans of young and sport-active adolescents.

## Figures and Tables

**Figure 1 ijms-24-01484-f001:**
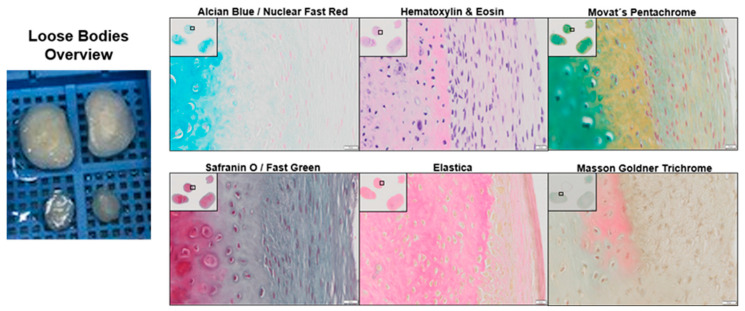
Histological staining of various regions from LB patient samples. In the top left corner are 4× overviews (total 40× magnification) and in the black square the selected regions for the 20× (total 200×) magnification. The scale bar equals 50 µm. For detailed explanation of the staining and the staining codes, see text Section 2.1 and Figure A1.

**Figure 2 ijms-24-01484-f002:**
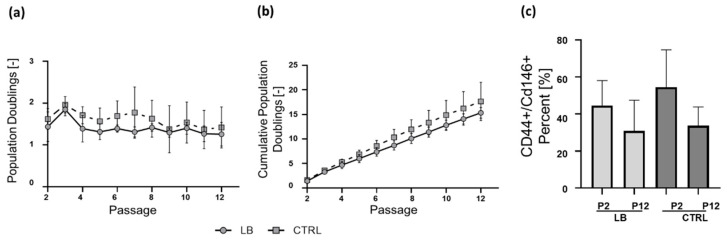
Long-term expansion and purity of LB cultures (**a**) Shown are the population doublings (PD) and (**b**) cumulative population doublings (CPD) for each indicated passage. Cells were sub-cultured in fixed seven-day intervals in T75 cm^2^ culture flasks from P2 to P12. Cell numbers were determined using a CASY^®^ cell counter (LB *n* = 7, CTRL *n* = 5, repeated measures, mean ± SD results, two-way ANOVA with multiple comparisons; Sidak). (**c**) Analysis of CDPC marker expression via flow cytometry. Shown are the percentage of double positive (CD44+ and CD146+) cells in early (P2) and late passage (P12) cultures. (LB *n* = 7, CTRL *n* = 5, one-way ANOVA with multiple comparisons; Tukey).

**Figure 3 ijms-24-01484-f003:**
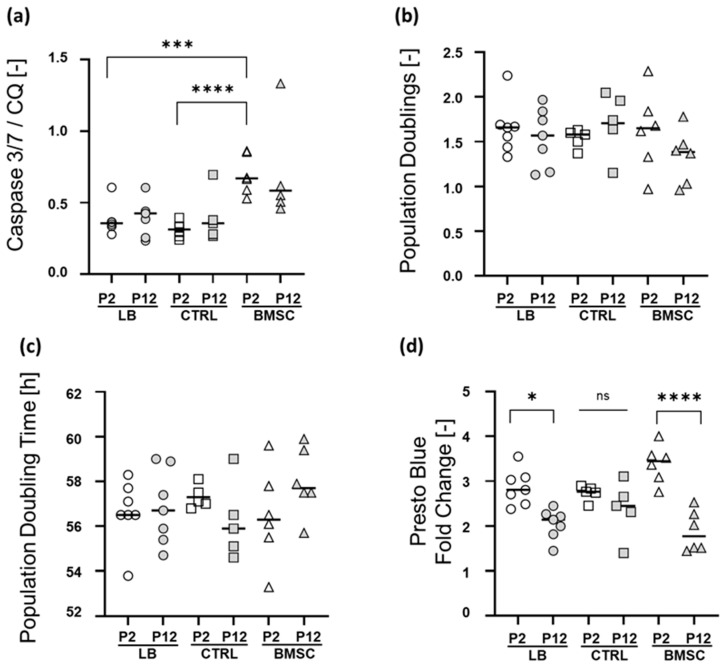
Apoptosis rate, short-term proliferation characteristic and metabolic activity. (**a**) Levels of active Caspase 3/7 as a surrogate marker of apoptosis were determined in early (P2) and late passage (P12) cultures of all three groups. Shown are the caspase 3/7 activities normalized to the cell number. (**b**) Population doublings (PD) [-] and (**c**) Population doubling time [h] of chondrocytes and BMSCs were determined over a nine-day culture period using CyQUANT™ cell proliferation assay. (**d**) The metabolic activity as an indicator for cell viability was determined using PrestoBlue™ viability assay over a nine-day culture period. Graph indicates the fold change in metabolic activity between day 0 and day 9. (LB *n* = 7, CTRL *n* = 5, BMSC *n* = 6, repeated measures, median line depicted, one-way ANOVA with multiple comparisons (Tukey), *p*-values of multiple comparisons as depicted; ns = not significant, * = *p* ≤ 0.05, *** = *p* ≤ 0.001, **** = *p* ≤ 0.0001).

**Figure 4 ijms-24-01484-f004:**
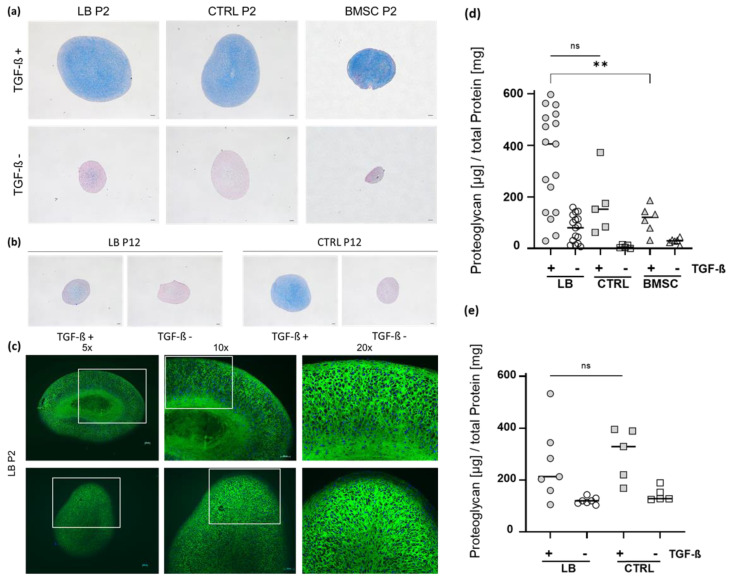
Chondrogenic differentiation of early and late passage chondrocytes and BMSCs. (**a**) Alcian blue staining of early passage (P2) LB-CH, CRTL-CH and BMSCs after 21 days of differentiation. Counter staining was performed with nuclear fast red (nucleic acid stain). All pictures are taken with a 5× magnification (total 50×), Scale bar: 100 µm. (**b**) Alcian blue staining of late passage (P12) LB-CH and CRTL-CH after 21 days of differentiation. Counter staining was performed with nuclear fast red (nucleic acid stain). (**c**) Immunohistochemistry of collagen type II (green) of early passage LB-CH. Nuclei were stained with DAPI (blue). (**d**) Proteoglycan content normalized to the total protein of early passage (P2) LB-CH (*n* = 17), CTRL-CH (*n* = 5), and BMSC (*n* = 6) cultures. (**e**) Smaller sample size was used to assess chondrogenic differentiation capacity of late passage (P12) LB-CH and CTRL-CH cultures; LB *n* = 7, CTRL *n* = 5. Repeated measures, median line depicted, one-way ANOVA with multiple comparisons (Tukey), *p*-values of multiple comparisons as depicted, ns = not significant, ** = *p* ≤ 0.01.

**Figure 5 ijms-24-01484-f005:**
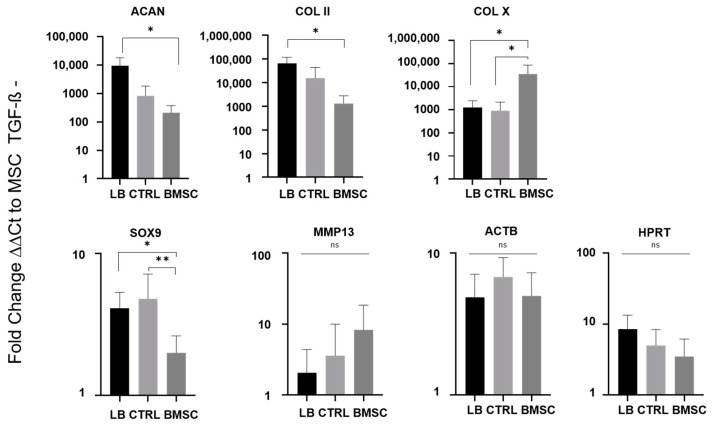
qPCR analysis of early passage cultures after chondrogenic (re-)differentiation. The mRNA expression of important components of chondrogenic phenotype and various classical housekeeping genes were analyzed 21 days after induction of chondrogenic differentiation. Evaluation revealed beta-2 microglobulin (b2 MG) as the most robust housekeeping genes for normalization (∆Ct) and BMSC cultures without TGF-ß supplement served as a reference sample (∆∆Ct). LB *n* = 17, CTRL *n* = 5, BMSC *n* = 6, repeated measures, one-way ANOVA with multiple comparisons (Tukey), *p*-values of multiple comparisons as depicted (ns not significant, * *p* ≤ 0.05, ** *p* ≤ 0.01).

**Table 1 ijms-24-01484-t001:** Demographic data and diagnostics. Summary of relevant information of the three sample groups.

Sample	Pat No	Source	Sex	Age	Diagnostics	Localisation	ICRS
**LB**	1	Knee	m	24	OD, microfracture and cancellous bone arthroplasty	MFC	III
	2		m	17	OD, cartilage defect 2 mm deep, width 7 mm	MFC	IV
	3		m	14	OD, 2.5 cm, MFC osteonecrosis	MFC	IV
	4		m	29	OD	MFC	III
	5		m	41	Acute left patellar dislocation with app. 1 × 1.5 cm cartilage flake distal patella	RP	III–IV
	6		m	27	OD, app. 2 × 4 cm	MFC	III–IV
	7		m	50	OD	MFC	IV
	8		f	49	OD, two app. 6 mm and 8 mm LBs ventral to eminentia intercondylaris	MFC	IV
	9		m	50	OD, app. 2 × 2.5 cm LB before anterior cruciate ligament; 1 × 1.5 cm LB slightly adherent	MFC	IV
	10		m	29	Trochlea cartilage damage, app. 2.5 × 2.5 cm LB right	LFC	IV a
	11		f	21	RP cartilage damage with loose body	RP	III a
	12		m	38	RP arthrosis right with LB, pangonarthrosis with corpora libra (III–IV), synovialitis right knee	RP	III–IV
	13		m	35	OD, defect app. 2 × 2 cm	MFC	III
	14		f	25	OD, app. 3 × 3 cm, 0.5 cm deep	MFC	IV b
	15		f	20	OD, LB app. 1 × 1.5 cm cartilaginous and bony (app. 2 mm layer; completely sclerosed)	MFC	IV
	16		f	20	OD, app. 2 × 2 cm	LFC	III
	17		m	32	OD, LB 1.5 × 2 cm, MFC damage, chondromalacia lateral tibial plaetau right	MFC	III–IV
	18		m	42	OD, app. 3 × 3 cm	MFC	IV
	*n* = 18		72% m/28% f	Mean 31.2 ± 11.2			
**CTRL**	19	Knee	f	75	Varus gonarthrosis		
	20		m	69	Varus gonarthrosis		
	21		m	53	Varus gonarthrosis		
	22		f	66	Posttraumatic gonarthrosis, secondary gonarthrosis		
	23		m	58	Varus gonarthrosis		
	*n* = 5		60% m/40% f	Mean 64.2 ± 7.8			
**BMSC**	24	Femural Head	m	88	Coxarthrosis		
	25		f	75	Coxarthrosis		
	26		m	42	Coxarthrosis		
	27		m	37	Post-traumatic coxarthrosis		
	28		m	74	Coxarthrosis		
	28		f	30	Dysplasia coxarthrosis		
	*n* = 6		67% m/33% f	Mean 57.7 ± 22.1			

Legend: **OD**, osteochondrosis dissecans; **ICRS**, International Cartilage Repair Society classification; **LFC**, lateral femoral condyle; **MFC**, medial femoral condyle; **RP,** retropatellar; app., approximately.

## Data Availability

The data presented in this study are available on request from the corresponding author. The data are not publicly available due to privacy and ethical reasons.

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
