# Peer review of "Chondrocyte Isolation from Loose Bodies—An Option for Reducing Donor Site Morbidity for Autologous Chondrocyte Implantation"

_ijms, 2023, doi:10.3390/ijms24021484_

Round 1
Reviewer 1 Report
In this article, authors investigate whether chondrocytes isolated from loose bodies could be used as potential source of cells for a subsequent autologous chondrocyte implantation.
As stated by the authors, the main limitation of the study is that Ctrl samples are obtained from different patients than loose bodies and for this reason results obtained from both samples are not comparable. Even more, mean age of patients from Ctrl cohort is 2 times higher than LB group. Authors affirm that taking healthy samples from the same patients does not comply with ethical criteria. What was going to be the subsequent treatment for these patients? If autologous chondrocyte implantation is the chosen option, the source of chondrocytes is healthy cartilage from a non-bearing area. Other treatment option would be to attach loose bodies to the subchondral bone or microfractures. Authors should clarify this point.
There are some publications that have found opposite results, i.e. Guillen-Garcia et al. Cartilage 2016, 149–156. In this article, authors reported that chondrocyte cultures can not be set up from all loose bodies. They found that chondrocytes can not be isolated from loose bodies neither completely detached from subchondral bone nor from patients with a disease of more than 1 year evolution. Could authors explain this fact?
Authors affirm that nutrition of detached cartilage is due to synovial fluid (lines 407-409) and this is the reason why there are some viable chondrocytes in LB. However, maybe subchondral bone has an also active role in feeding the cartilage since the surgical attachment of loose bodies not always works and patients with OCD frequently have to undergone a cartilage repair procedure.
Statiscal methods should be reported in an appropriate section in Material and Methods
Lines 272-273: Means seem to be comparable but it would be useful to have a p-value also
Author Response
Dear editor, dear reviewer:
Thank you for the thorough review of our manuscript. We appreciate the propositions provided and we addressed all questions point by point below. All changes within the manuscript were done in ‘track mode’. Please find our response to the reviewer requests point by point below:
Reviewer 1:
- As stated by the authors, the main limitation of the study is that Ctrl samples are obtained from different patients than loose bodies and for this reason results obtained from both samples are not comparable. Even more, mean age of patients from Ctrl cohort is 2 times higher than LB group. Authors affirm that taking healthy samples from the same patients does not comply with ethical criteria. What was going to be the subsequent treatment for these patients? If autologous chondrocyte implantation is the chosen option, the source of chondrocytes is healthy cartilage from a non-bearing area. Other treatment option would be to attach loose bodies to the subchondral bone or microfractures. Authors should clarify this point.
In addition to the LOOSE BODIES, healthy cartilage was also harvested from the non-weight-bearing area of the patients in the LB group. From these biopsies, chondrocytes were isolated for standard ACI under GMP conditions. All obtained cells were subsequently transplanted according to routine protocol and could not be used for our research. The aim of our work was to circumvent the collection of cartilage biopsies from intact (presumably healthy) non-weight bearing areas of the knee joint by using cells from the LB. Of course, there are other treatment options for these patients (e.g., microfractures etc.), but various clinical trials (including phase III) show that ACI is superior to these procedures.
Corresponding statement is now included and reads as follows:
“Although appropriate biopsies for standard ACI were also taken from the patients in the LB group and cells were isolated under GMP conditions, we could not include them in our research as they were completely required for the treatment according to the routine pro-tocol. The aim of our work was to circumvent the collection of cartilage biopsies from in-tact (presumably healthy) non-loaded areas of the knee joint by using cells from the LB. Other treatment options are of course available for these patients (e.g., microfracture, etc.), but several clinical trials (including phase III) show that ACI is superior to these alterna-tive procedures [43].”
- There are some publications that have found opposite results, i.e. Guillen-Garcia et al. Cartilage 2016, 149–156. In this article, authors reported that chondrocyte cultures can not be set up from all loose bodies. They found that chondrocytes can not be isolated from loose bodies neither completely detached from subchondral bone nor from patients with a disease of more than 1 year evolution. Could authors explain this fact?
In our study, from all samples, we were able to establish cell cultures. In rare cases we had to discard samples due to early contamination. In many studies as well as in the described paper from Guillen-Garcia et al. the researchers opted for an extensive overnight digestion, which to our experience can lead to a reduced viability. Furthermore, Guillen-Garcia et al. filtered the cell solution afterwards, which can also damage or disrupt cells by shear forces particularly further after overnight digestion. In this study, also only 32% of the patients showed a classical OCD; 13 patients were arthritic (chronical inflammation) and 10 trauma patients, so these samples differ from OCD samples. All these factors might explain the low rate of established cell cultures.
We have included the following statement regarding this effect in the text:
“Some other publications report that it is not possible to establish chondrocyte cultures from all LBs, especially in patients whose disease onset dates back more than one year [42]. We were able to isolate chondrocytes from all LBs obtained and only in two exceptional cases had to discard isolated cells because of early bacterial contamination. Com-pared with chondrocytes from healthy cartilage regions, the isolated cells showed an excellent proliferation rate, cartilage marker expression, and chondrogenic matrix production and were even superior to human BMSCs. The reason for the difference between our results and those of others may be that a longer overnight digestion was chosen in these studies, which in our experience may lead to decreased viability [42]. In addition, the isolated cell solutions are often subsequently filtered, which may also lead to damage or destruction of the cells by shear forces, especially after the prolonged digestion process.”
- Authors affirm that nutrition of detached cartilage is due to synovial fluid (lines 407-409) and this is the reason why there are some viable chondrocytes in LB. However, maybe subchondral bone has an also active role in feeding the cartilage since the surgical attachment of loose bodies not always works and patients with OCD frequently have to undergone a cartilage repair procedure.
To address this comment, the following correction has been done: “Even after longer time periods of detachment in vivo, the survival and supply of nutritients for LBs is archived via the synovial fluid [38] and actively by the subchondral bone as shown in animal studies [39,40].”
- Statistical methods should be reported in an appropriate section in Material and Methods
Corrected accordingly. Section “2.12. Statistical Analysis” has been added in the revised manuscript:
“2.12. Statistical Analysis
All statistical analysis was performed within the software Prism 9.41 (Dotmatics). One-way ANOVA with multiple comparisons and Tukey compensation was utilized for all analysis, except for the CPD analysis, which was performed using a two-way ANOVA multiple comparison and Sidak compensation for more robustness.”
- Lines 272-273: Means seem to be comparable but it would be useful to have a p-value also
Line 272-273 refers to the histological analysis results without mentioning p-values. Otherwise, p-values are given where applicable.
We hope that all aspects of the minor revision are sufficiently addressed.
Yours sincerely
- Textor, S. Geissler & team

Reviewer 2 Report
In the article: “Chondrocyte isolation from loose bodies – An option for reducing donor site morbidity for autologous chondrocyte implantation”, the authors discussed the potential isolation of chondrocytes form loose body region and their application for autologous chondrocyte.
Overall, this manuscript results very interesting, the authors clearly explain the rational of the study and discussed the topic point by point.
However, we would like to invite the authors to clarify some minor points:
1. Please check the check punctuation and spaces;
2. Page 2, lines 63-64: the authors introduce the cartilage disease, please spend some words about the features of this pathology;
3. Page 3, lines 126-127: the authors said “Therefore, isolated cartilage was cut into 2 x 4 mm fragments and digested sequentially”. How can you be so accurate with the measurements? Please, specify the procedure or delete this detail if you are not sure if it is correct;
4. What is the concentration of Collagenase II used for cartilage digestion? Please, specify among materials and methods section;
5. After the isolation, did you check the correct phenotype of chondrocytes? Did you verify the expression of specific phenotype biomarkers? Some pictures of isolated cells are available?
6. Page 3, lines 137-138: the authors said “All assays were carried out with primary cells either in passages 2 – 4 or 10 – 12”. How is it possible to reach such high passages with primary cells? is it known that they go into senescence or do they differentiate into fibroblasts. Are there photos for each step? Have phenotype markers been verified?
7. Why MMP-13 gene expression was evaluated? What does mean?
8. Protein expression/secretion of specific biomarkers was considered? Western blotting? ELISA?
9. Please, describe in a better way the normalization of qRT-PCR data;
Author Response
Dear editor, dear reviewer:
Thank you for the thorough review of our manuscript. We appreciate the propositions provided and we addressed all questions point by point below. All changes within the manuscript were done in ‘track mode’. Please find our response to the reviewer requests point by point below:
Reviewer 2:
- Please, check the check punctuation and spaces;
We rechecked all punctuations and spacing and corrected all errors.
- Page 2, lines 63-64: the authors introduce the cartilage disease, please spend some words about the features of this pathology;
As requested, we added a short description in line 63 and Ref [1]: “The most common disorder in which loose bodies occur is osteochondritis dissecans (OCD). It might result most commonly in pain, predominantly in late stages, due to physical activities. Other symptoms might include swelling, joint locking or decrease in the range of motion. Overall, there are no general typical clinical signs [1].”
- Page 3, lines 126-127: the authors said, “Therefore, isolated cartilage was cut into 2 x 4 mm fragments and digested sequentially”. How can you be so accurate with the measurements? Please, specify the procedure or delete this detail if you are not sure if it is correct;
This criticism is valid - not all fragments had the exactly same size, it was an approximation. We changed the statement (nor line 135) to “…cut into small fragments of approximately 2 – 4 mm diameter…”.
- What is the concentration of Collagenase II used for cartilage digestion? Please, specify among materials and methods section;
Concentrations have been added: “…first using 350 U/ml pronase E (Sigma Aldrich, USA) and then 100 U/ml collagenase type II (Biochrom, Germany)…”
- After the isolation, did you check the correct phenotype of chondrocytes? Did you verify the expression of specific phenotype biomarkers? Some pictures of isolated cells are available?
All isolated cells showed the same microscopic phenotype. Some cell culture pictures were added as Appendix A2 in the new version of the manuscript. In short, the shape of the isolated cells was that of dedifferentiated chondrocytes small, spindle-shaped form or fibroblast-like [see also doi: 10.1385/1-59259-861-7:069]. Their chondrocyte nature was confirmed via 3D chondrogenic spheroid differentiation. We tested only for the reported surface marker.
- Page 3, lines 137-138: the authors said “All assays were carried out with primary cells either in passages 2 – 4 or 10 – 12”. How is it possible to reach such high passages with primary cells? is it known that they go into senescence or do they differentiate into fibroblasts. Are there photos for each step? Have phenotype markers been verified?
We were ourselves very surprised but did not detect cell cycle arrest, although population doublings declined. We provided our isolation and culture protocol in detail to aid reproducibility as this result may be due to technical steps as well as the selection of superior FBS.
- Why MMP-13 gene expression was evaluated? What does mean?
MMP13 plays a central role in extracellular matrix turnover and transient cartilage. Therefore, it is also relevant in the 3D spheroid differentiation process as a quality marker (the lower the better).
- Protein expression/secretion of specific biomarkers was considered? Western blotting? ELISA?
We thought about ELISA but focused more on the re-differentiation analysis. In addition, the protein yield after the differentiation was not sufficient for Western blots.
- Please, describe in a better way the normalization of qRT-PCR data;
In line 203ff, we added a more detailed explanation of the normalization process: “Normalization was conducted by relating the genes to the housekeeping gene beta-2 microglobulin (b2MG) yielding ∆Ct. Afterwards each gene was normalized to the corresponding gene of the TGF-ß-negative BMSC culture resulting in the ∆∆Ct.”
We hope that all aspects of the minor revision are sufficiently addressed.
Yours sincerely
Textor, S. Geissler & team

Round 2
Reviewer 1 Report
Authors have completely addressed all the questions and the manuscript can be accepted for publication
Reviewer 2 Report
now the paper is suitable for pubblication